# A Method to Reuse Archived H&E Stained Histology Slides for a Multiplex Protein Biomarker Analysis

**DOI:** 10.3390/mps2040086

**Published:** 2019-11-15

**Authors:** James P. Hinton, Katerina Dvorak, Esteban Roberts, Wendy J. French, Jon C. Grubbs, Anne E. Cress, Hina A. Tiwari, Raymond B. Nagle

**Affiliations:** 1Cancer Biology Graduate Interdisciplinary Program, University of Arizona Cancer Center, Tucson, AZ 85724, USA; jhinton@email.arizona.edu; 2Ventana/Roche Tissue Diagnostics, Tucson, AZ 85755, USA; katerina.dvorak@roche.com (K.D.); esteban.roberts@roche.com (E.R.); wendy.french@roche.com (W.J.F.); jon.grubbs@roche.com (J.C.G.); 3Department of Cellular and Molecular Medicine, University of Arizona, Tucson, AZ 85724, USA; 4Department of Medical Imaging, College of Medicine, University of Arizona, Tucson, AZ 85724, USA; hinaarif@radiology.arizona.edu; 5Department of Pathology, College of Medicine, the University of Arizona, Tucson, AZ 85724, USA; rnagle@email.arizona.edu

**Keywords:** immunohistochemistry, H&E stained slides, de-stain, re-stain, image analysis

## Abstract

Archived Hematoxylin and Eosin (H&E) stained pathology slides are routinely stored to index formalin-fixed paraffin-embedded (FFPE) sample tissue blocks. FFPE blocks are clinically annotated human tumor specimens that can be valuable in studies decades after the tissue is collected. If stored properly, they have the potential to yield a valuable number of serial sectioned slides for diagnostic or research purposes. However, some retrospective studies are limited in scope because the tissue samples have been depleted or not enough material is available in stored blocks for serial sections. The goal of these studies was to determine if archived H&E-stained slides can be directly reutilized by optimizing methods to de-stain and then re-stain the H&E stained slides to allow the detection of several biomarkers of interest using a conjugated antibody with chromogen multiplex immunohistochemistry procedure. This simple but innovative procedure, combined with image analysis techniques, demonstrates the ability to perform precise detection of relevant markers correlated to disease progression in initially identified tumor regions in tissue. This may add clinical value in retaining H&E slides for further use.

## 1. Introduction

In immunohistochemistry, several types of tissue immunostains are utilized to analyze morphological features, cellular structures, cell type, and the presence or absence of microorganisms. The most popular of the staining methods for diagnostic potential is the utilization of hematoxylin and eosin (H&E) staining [1]. H&E stains reveal structural information, with specific functional implications. H&E staining of tissue is used to assess cellular and morphological structures, identify the type of tissue, morphological variability, cell type, and pathological changes. The use of H&E staining has been the most effective and utilized procedure for pathological diagnosis of patient neoplasia for over a century [2,3]. It has allowed pathologists to pinpoint focal areas of a specimen-containing aggressive tissue and foster a proper diagnosis [4]. Therefore, developing procedures to re-utilize these archived samples to determine individual biomarker expression levels (and potential protein–protein association) could assist in determining disease progression and directions for appropriate treatments.

H&E staining is used in conjunction with a variety of tissue fixatives and allows the display of various cellular and tissue components, including the extracellular matrix, the cellular cytoplasm, and the nuclear structures [3]. The hematoxylin is converted into its oxidization product hematein, which is a basic dye that stains acidic (basophilic) tissue components (ribosomes, nuclei, and rough endoplasmic reticulum) a darker purple color and the acidic eosin dye stains other protein structures of the tissue (stroma, cytoplasm, muscle fibers) a pink color [2,4,5]. They are also valuable in distinguishing normal structural components from neoplastic regions. However, with the current procedures, H&E staining is utilized along with sequential sections stained with antibody. Serial sectioning may cut through the region of intent and may result in the loss of regions necessary for critical diagnosis. This is particularly an issue with smaller core needle biopsies (CNBs) that are of a limited size and number. These samples are considered “precious” in regard to availability and require the utmost accuracy in testing procedures to result in proper diagnoses.

A major advantage of a method that allows the reuse of the H&E-stained slide is that it will alleviate the need for additional sequentially sectioned slides, particularly with the diminutive CNBs. Due to the of size of CNBs, they are also subject to tissue sample exhaustion with the loss of the diagnostic lesion. This method would present a major practicality when a particular region of interest is no longer available in the sample block due to sequential cuts. The ability to reuse the initial H&E containing the lesion could be critical. De-staining these H&E-stained tissue slides could also potentially reduce the need for re-biopsy.

Another advantage of re-staining archived H&E-stained slides is due the rapidly expanding use of whole-slide imaging (WSI), also known as digital pathology (DP) or virtual pathology. DP is a technology that involves the high-speed and high-resolution digital acquisition of images representing entire stained tissue sections from glass slides in a format that allows them to be viewed by a pathologist on a computer monitor [6]. This streamlines the ability of a surgical pathologist to make a primary diagnosis utilizing digitized images of the H&E-stained slide, allowing digital preservation while the H&E and other stains are fresh [7]. As the validation of this technology becomes widespread, the method reported here could be used for analysis of stored H&E-stained slides for subsequent diagnosis of tumor subtypes within a patient sample or future discovery of novel target proteins.

## 2. Experimental Design

This procedure details steps to de-stain and reutilize archived H&E stained slides for antibody immunostaining modalities. For our research, prostate cancer was initially chosen due to frequent limitations of tissue in sample biopsies and the requirement for biomarker study. Prostate cancer (PCa) is also known to express variable levels of several markers associated with disease progression, such as phosphatase tensin homolog (PTEN) and ETS-Related Gene (ERG), making it a viable target for testing this procedure. A link between the PTEN pathway and ERG protein expression has previously been evaluated in various prostate cancer studies [8,9,10,11,12,13]. In studies investigating the trend of PTEN loss in tumors of prostatectomies and locally recurring castrate-resistant prostate cancers (CRCPs) with ERG overexpression, the data showed that the loss of PTEN was significantly associated with ERG positivity [11]. Another study indicated that the combination of ERG overexpression and PTEN deletion is common in aggressive capsular penetrating lesions [14]. Therefore, we decided that using antibodies targeting PTEN and ERG would be the validated markers in this study.

We first used archived H&E-stained slides from PCa resections or biopsies stored for at least one year (with film coverslips) to demonstrate proof that the H&Es could be reutilized for biomarker stain using an H&E de-staining procedure with standard laboratory equipment and reagents. De-identified patient tissue samples were provided with no link to information that can be used to identify patients. Oversight for tissue acquisition was managed by the UA Cancer Center, an NCI designated Comprehensive Cancer Center. De-identified FFPE prostate tissue multi-Array (TMA), adenocarcinoma, lung, colon, and skin tissue slide samples (Table 1) were acquired from Roche Tissue Diagnostics (RTD)/Ventana Medical Systems Inc. (VMSI or Ventana), Tucson, Arizona. De-identified PCa CNBs (Table 1) used for initial feasibility testing of these procedures, were provided and serial sectioned by the University of Arizona Cancer Center Tissue Acquisition and Cellular/Molecular Analysis Resource (TACMASR) core support service with the approval of the University of Arizona Institutional Review Board (IRB). Initial H&E analysis (Table 1), was provided by Dr. Ray Nagle. Several of the additional sample tissue specimens used within this study (liver, lung, normal colon and PCa TMA) were exhausted and only the H&E stained slide remained available for testing. For the samples in the cohort that retained tissue availability, sample slides for each tissue were H&E-stained and cover-slipped using the reagents and staining procedures on the Sakura Tissue-Tek Prisma *Plus* & Film Automated Slide Stainer & Coverslipper at VMSI and allowed to air dry in a fume hood for approximately 10 min.

### 2.1. Reagents and Materials

Acetone (VWR, Visalia, CA, USA; BDH1101-4LP)Xylene (Millipore, Billerica, MA, USA; 58235)100%–95% EthanolDistilled water (DI)Reaction Buffer (Tris based, 7.6 ± 0.2 pH) (Proprietary reagent, Ventana/RTD, Tucson, AZ, USA)H&E stained tissue slidesParaffin embedded sample tissuesMicroscope slides (Matsunami TOMO^®^, Bellingham, WA or Superfrost™, Thermo Fischer Scientific, Waltham, MA, USA)Slide coverslips (VWR, Visalia, CA, USA; 48393 251)Mounting medium (coverslip sealant) (Thermo Scientific, Waltham, MA, USA; 8312-4)

### 2.2. Equipment

Slide Baskets (Sakura Finetek, Torrence, CA, USA; 4768)Plastic staining dish (container) [with lids preferably] (Sakura Finetek, Torrence, CA, USA)ForcepsKimwipes™Brightfield microscope (Olympus BX40)Parafilm (optional) ((Parafilm M^®^ Laboratory film, VWR, Visalia, CA, USA)Slide Scanner (Leica Aperio AT2; Leica Biosystems, Buffalo Grove, IL, USA, DP200 Slide Scanner RTD, Tucson, AZ, USA)Incubator (60 °C ± 5 °C) (VWR, Visalia, CA, USA)BenchMark ULTRA (optional)Slide Coverslipper (Sakura Tissue -Tek Prisma Plus & Film Automated Slide stainer and Coverslipper; Sakura Finetek, Torrence, CA, USA)

The initial antibodies chosen for the proof-of-concept testing were the on-market products VENTANA anti-p40 (B28) mouse monoclonal antibody (data not shown), anti-cytokeratin 5/14 (CK5/14) (EP1601Y/LL002) rabbit and mouse monoclonal antibody cocktail from Cell Marque (Figure 1) and a rabbit polyclonal antibody (Ab) against the laminin-binding extracellular domain of integrin alpha 6 (CD49f) from the lab of Dr. Anne Cress at the University of Arizona, Department of Cellular and Molecular Medicine. The CD49f antibody was formulated and optimized from a 1 mg/mL stock concentrate to a 1:800 dilution in a pH = 7.3 Phosphate Avidin antibody diluent containing a proprietary B5 blocker, goat globulins, and 55 mM NaCl concentration with Proline preservative.

The antibodies utilized for additional immunostaining of five selected de-stained archived H&E stained slides to analyze and compare immunostaining intensities with corresponding sequential slides were the VENTANA anti-High Molecular Weight Cytokeratin (HMWCK) and p63 (34βE12 + 4A4, respectively) mouse basal cell cocktail, Cell Marque anti-CK 8&18 (B22.1 &23.1) rabbit monoclonal, VENTANA anti-E-cadherin (36) mouse monoclonal, anti-CD49f and VENTANA anti-ERG (EPR3864) rabbit monoclonal. Each reused (re-stained) H&E stained slide was de-stained according to the procedure listed in this report and immunostained with the various antibodies along with the corresponding sequential slide for each sample (see Results section).

IHC DAB detection was accomplished by utilizing a VENTANA OptiView DAB IHC Detection kit and VENTANA *ultra*View DAB IHC Detection Kits. Chromogen detection was accomplished by utilizing a VENTANA Discovery Chromomap Detection Kit to target and detect the anti-HMWCK + p63 mouse monoclonal antibody cocktail with the secondary antibody linker conjugated with hydroxyquinazoline and anti-hydroxyquinazoline (HQ) horseradish peroxidase (HRP) enzymes for detection with a VENTANA Discovery Purple Chromogen. The detection for the anti-CD49f rabbit polyclonal antibody included an anti-rabbit nitrolpirazole (NP) conjugated secondary and anti-NP HRP for detection of the VENTANA Discovery Teal Chromogen. The detection of the sample slides was accomplished utilizing the LEICA AT2 slide scanner and the VENTANA Digital Pathology 200 (DP200) slide scanner.

## 3. Procedure

### De-Staining the H&E Slides. Time to Completion: ~2 h

Perform the de-staining procedures in a fume hood using manual wash stations (baths) containing the reagent solvents listed in reagents and materials section. These procedures negate the need for heat to remove sealed coverslips and result in safe removal without tissue damage.

1.Place the H&E archived index slides into slide baskets to allow manual rinsing.2.Manually soak slides in Acetone for 10 min to remove coverslip.


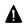
**CRITICAL STEP** Cover the container of acetone with either container lid or parafilm to reduce acetone evaporation (evaporation reduce effectiveness of coverslip removal).

3.Remove the coverslip with forceps slowly to reduce damage to tissue (longer intervals allow coverslip to slide off independently). Discard coverslip.4.Moderately rinse slides 3 times in xylene bath to remove any remaining adhesive, allow slides to sit in the bath to remove all sealant (Approximately 1-min hold times should suffice between rinses). [Rinse = 30+ Dips in reagent] [Hold = allowing slides to sit in reagent container between rinses].5.Rinse slides (5–6 times) with 3 min hold intervals between rinses in 95% EtOH for ~30 min to remove eosin stain. [Total time for 95% EtOH procedure is approx. 30 min, however intervals may be increased for removal of eosin from larger tissue sections.]


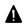
**CRITICAL STEP** Cover with caps or paraffin between rinses to reduce reagent evaporation. Moderately rinse slides during hold times to expedite eosin removal.

6.Rinse slides in distilled (DI) H_2_O and lightly tap on Kimwipes™ to remove excess.7.Rinse slides in Reaction Buffer (3–4 rinses with 2 min hold intervals) to remove Hematoxylin (cap or paraffin cover to reduce evaporation).8.**OPTIONAL STEP** Apply extra rinses in reaction buffer to expedite removal of hematoxylin.9.Allow slides to dry in hood in fume hood for 5 min (do not allow tissue to dry completely this will impact immunostaining intensity).10.Run antibody IHC assay detection protocols on instruments or manual procedures (will be variable, depending on biomarker).11.Wash completed slides in water and dawn dish detergent mix to remove any residual reagents12.Dehydrate slides and coverslip on Sakura coverslipper or manual dehydrate rinse with xylene > acetone > 80% EtOH > 90% EtOH > 100% EtOH > xylene and coverslip using mounting medium.


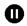
**PAUSE STEP** Slides may remain in xylene and reaction buffer steps for extended periods without damaging the tissue.

## 4. Expected Results

The method described in this study utilized forty-nine sample H&E-stained resection and CNB slides that were analyzed and commented on by a board-certified pathologist for normal or neoplastic status, Gleason grade, preservation status, and any distinguishing features for the categorization of potential aggressiveness. The initial testing was accomplished using DAB IHC detection kits to determine retention of marker stain intensity. During the initial stages of this study, multiple test samples demonstrated lower intensities as a result of utilizing an un-optimized protocol (data not shown). However, continued editing and updates to the initial procedure on re-utilized H&E index slides resulted in viable stain intensity, demonstrating the feasibility of the procedure and potential for optimization to culminate in stain intensity comparable to that of sequential slides utilizing standard procedures (Figure 2).

The initial testing procedures resulted in moderate but visible stain intensity providing proof-of-concept. At this stage, further optimization and repeat testing was warranted to increase the stain intensity to comparable levels of those occurring using the standard antibody staining methods and to ensure reproducibility. The procedure methods were improved by four steps: 1. Applying timed reagent rinse procedures at the xylene, ethanol (EtOH), and Ventana/RTD proprietary reaction buffer steps (1-min rinse times between each hold); 2. Increasing EtOH and reaction buffer reagent rinses from 1 rinse to 5–6 and 3–4 manual rinses respectively for optimal efficiency; 3. Including an approximate 5-min drying step after the reaction buffer rinse to limit residual excess reagent interference in the online application of biomarkers; and 4. Editing online cell conditioning steps (for heat induced antigen retrieval) to reduce potential epitope destruction.

This procedure optimization was considered the standard when applied to any H&E-stained slide stored for up to 2 years but needed further optimization for tissues stored for periods 2 years or longer. The subsequent experimentation steps employed the use of antibodies targeting PTEN and ERG biomarkers, VENTANA anti-PTEN (SP218) mouse monoclonal antibody, and anti-ERG (EPR3864) rabbit monoclonal antibody. These validated markers were used since they demonstrate 1. the heterogeneous variability of aggressive prostate cancer and 2. the comparative expression of PTEN loss and ERG expression in aggressive tumors. In this procedure, steps (1–8) represent the optimized H&E de-staining procedures. However, during the testing, unforeseen scheduling resulted in slight deviations (extended reagent HOLD times) in steps 4 and 7 that lead to the determination that certain steps, which were the xylene and reaction buffer HOLD times, could be amended without incurring damage to samples. The updated procedure, which only involved an extended xylene hold time and is essentially the same optimized procedure, resulted in comparable stain intensity to the standard protocol and allowed the ability of distinct determination of aggressive tumor areas (Figure 3). After the successful completion of a sequential round of experimentation using IHC DAB, we tested whether antibodies targeting multiple biomarkers could be applied for detection with the use of chromogenic detection reagents. Again, prostate adenocarcinoma CNBs were used as experimental specimens for the de-stain and re-stain procedure. The antibodies chosen were specific for the integrin α6 (CD49f) laminin-binding domain and HMWCK + p63. These markers were chosen due to known membranous expression levels (CD49f) and cytoplasmic and nuclear (HMWCK + p63 respectively) positive expression levels in non-neoplastic basal cells of prostate tissue. In PCa, CD49f expression is membranous and aggressive and invasive disease exhibits an intracellular expression pattern [14,15,16,17,18]. It is also associated with poor patient prognosis, reduced survival, and increased metastasis [19,20,21]. These markers were not expected to colocalize but to demonstrate the expression pattern of both non-neoplastic and neoplastic structures and focal areas in tissue after marker application, following the de-stain procedure.

The expected outcome was to demonstrate definitive areas of non-neoplastic vs. tumor regions with the application of antibodies and chromogen detection. This would allow the simultaneous detection of normal and aggressive structures in one tissue sample after pathologist analysis of the H&E-stained slide, allowing for the potential utilization of one slide. The results demonstrate strong stain intensities for both targets and well-defined areas of demarcation of non-neoplastic vs. tumor structures. As expected, both markers are visible in normal basal cells of normal prostate glands (although the HMWCK + p63 stain intensity primarily masks the CD49f signal in those areas), but CD49f antibody clearly displays an intracellular and cytoplasmic expression in the areas of budding tumor (Figure 4).

These positive results from testing samples archived up to 2 years warranted the evaluation of the potential ability of this procedure to be utilized with other tissue types, for H&E stained slides archived 2 years or more, and samples archived utilizing glass coverslips. Therefore, five archived PCa CNBs (2 years 11 months), a normal colon (2 years 1 month), liver and lung samples (4 years) along with 4-plus year (4+) PCa resection (4 years 11 months) H&E stained sample slides sealed with thin film coverslip were tested. For the testing of H&E stained slides sealed with glass coverslips, archived PCa CNBs (2 years 11 months), skin samples (5 years) and a PCa TMA sample (12 years) were tested. During the execution of this procedure, the removal of the coverslip was determined to be a limiting factor. Therefore, the parameters involved with the removal was tracked and recorded in this report (Table 2). After the removal of the H&E stain, the sample slides were re-stained with selected antibodies utilizing optimized protocols adapted for IHC on de-stained H&E slides (Table 3).

The testing of the H&E stained slide samples archived 2 years or more involved extended coverslip removal and reagent rinse times (2+, 4, 4+, 5, and 12-year archived samples) which indicated that archival time, storage condition and coverslip type may play a factor in slide processing with the procedure. The processing of 2-year archived H&E stained slides sealed with thin film (all PCa CNBs) only required minimal extension time of coverslip removal (to ~60 min) but resulted in H&E stain removal and comparable antibody immunostaining intensities compared to the corresponding sequential slides (Figure 5A–6E). The reused H&E and corresponding sequential slides were evaluated by a board-certified pathologist in a side by side comparison for immunostaining intensity (Table 4). The histopathologic analysis focused on any present tumor or normal regions for intensity. The data analysis indicates that there was a significant matching in the immunostaining intensities between the reused H&E stained slides and sequential comparator slides immunostained with the various antibodies (Figure 5F).

The processing of the 4 and 4+ year archived H&E stained slides (PCa resections) sealed with thin film coverslip required extended time of coverslip removal (~38 and 47 h) but resulted in H&E stain removal. The archived H&E stained slides sealed with glass coverslip for 2+, 5 and 12-year (PCa resection, two skin and PCa TMA) required 1–2 days and 4–5 days for coverslip removal. The 4+ year archived sample resulted in comparable intensity (CK 8 &18) to the sequential comparator slide (Figure 6). The reused 12-year archived PCa TMA H&E stained slide immunostained with Ventana anti-ERG resulted in immunostaining but intensity was variable across different cores but demonstrated feasibility (data not shown). The reused 5-year archived H&E stained slides (skin resections) sealed with glass coverslips also required extended time for coverslip removal and reagent rinses. The resulting H&E stain removal exhibited residual H&E stain on the slides resulting in incomplete immunostaining (ERG) (data not shown). The resulting retention of the hematoxylin and eosin on the slides may have potentially impacted the results and will need further inquiry on storage conditions to ascertain steps to mitigate any issue. We found that the storage conditions of older H&E stained slides (particularly with glass coverslips) caused extensive adhesion of the coverslip to the tissue slide due to the extended time in storage, requiring a slight extension of extraction procedures. In addition, we observed that pre-analytics impacted H&E removal, resulting in some residual retention. Unfortunately, due to age of the slide, pre-analytical data was not available.

## 5. Discussion

When patients are suspected of having PCa, a tissue sample is required for diagnosis. The sampling of the potentially neoplastic area may be assisted through means of ultrasound (US) or multiparametric magnetic resonance imaging (mpMRI) guided techniques. Sample resections and needle biopsies are routinely formalin-fixed and processed and embedded for histological sampling then stained for H&E and IHC, allowing pathologists to analyze an excised patient tissue sample from the affected area after diagnosis to differentiate between cancer and non-neoplastic events, such as benign prostatic hyperplasia. The H&E-stained slide plays a critical role in assisting the diagnosis of the pathologist in corroborating the initial findings with MRI and US procedures. Traditionally, after pathologist analysis and diagnosis, the samples can then be processed with biomarkers targeting detection of epitopes that are overexpressed in aggressive tumors. Currently this is the standard procedure deployed in companion diagnostics that allows for the stratification of patients who may benefit from a specific therapeutic intervention.

The accurate evaluation of biomarkers with these samples is critical for patient diagnosis, particularly with smaller samples, such as CNBs, fine needle aspirates, and potentially transurethral resection of prostate samples (TURPS). The smaller size of these tissue samples limits tissue availability and requires precise testing for important results. Loss of available tissue slides is a risk that could be mitigated with the use of H&E slide de-stain and re-stain procedures. The potential to detect multiple markers using chromogenic multiplexing on a single indexed tissue slide that had been analyzed and diagnosed by a pathologist to definitively contain aggressive tumor, leaves open the possibility of predictive companion diagnostics with minimal sampling. This may provide the opportunity for a one sample/one result diagnosis limiting the invasive nature of tissue specimen collection, which benefits the patient greatly.

There are few reports that provide instructions for removal of the H&E staining that leaves the target epitopes intact for potential reuse of the slide for selective biomarkers. Current existing protocols (and forums) only discuss de-stain procedures for slides that have stained inadequately, or have been stained with excessive hematoxylin and have lengthy protocol steps that may extend the procedure hours to days. Others may require the use of more corrosive reagents (% HCL solutions). Procedures utilizing either beta-mercaptoethanol/sodium dodecyl sulfate (2ME/SDS), 6 guanidinium hydrochloride (GnHCL) or 6 M Urea have been demonstrated to elute antibodies from immunostained tissues on positively charged glass slides (or glass coverslips) for sequential antibody re-stain [22,23]. However, these methods focused on the removal of the bound primary antibody and the reagents used were not intended to remove the H&E stain. For this report, an innovative method utilizing non-corrosive reagents was created and applied in a particular procedure using these reagents in sequence that optimized the H&E slide de-stain. This procedure removed the majority of the visible stain while retaining tissue integrity and morphology and allowed the preparation of specified IHC protocol to re-stain the sample sides. The primary tissue sample used for initial testing was prostate adenocarcinoma, however, this will translate to other tissues.

This study utilized liver, colon, skin and PCa resections and CNB samples for procedure testing. The study included the addition of antibodies detecting clinically relevant biomarkers such as PTEN, ERG, E-cadherin, Racemase (p504s), cytokeratin 8 and 18 and the CD49f protein for potential indication of aggressiveness and antibodies against HMWCK cocktailed with a p63 marker (a p53 homologue containing the N-terminal transactivation domain) as well as the variant p40 marker (lacking the N-terminal domain), that will detect the presence of normal basal cells of prostatic glands. These antibodies were critical in detection of differentiating prostatic adenocarcinomas vs. the detection of non-neoplastic prostatic tissue, as well as determination of intracellular marker activity and basal cell attenuation, respectively. Moreover, during this study the positive outcome from testing various tissue samples archived beyond 4 years utilizing thin film and 12 years with glass coverslips yielded promising results indicating tissue epitopes remain stable on H&E stained slides archived at a minimum of 4 years. This indicates that the procedure may be useful for the interrogation of other clinically relevant proteins in tissues other than prostate and for H&E-stained slides stored for longer periods of time. However, the conditions of the slide storage and the type of adhesives applied to seal the slide may have an impact on results. Another factor may be the specific antibody selected for each specific study. The antibodies used for this study yielded promising results but each antibody demonstrates various qualities, therefore, continued optimization may be warranted for this procedure.

Further experimentation will be repeated involving archived specimen slides utilizing film coverslips that have been stored for 4 years and more, as well as continued interrogation of samples sealed with glass coverslips. This will determine the robustness of the procedure to encompass reproducible testing of samples from decades past to incorporate newly discovered targets to test protein expression that may offer answers to questions that may have remained unsolved. Moreover, with the development of newer chromogen dyes, the possibility of utilizing one slide for multiple markers may now become a distinct possibility saving valuable time and resources. The results demonstrated in this report can be considered the first step towards a more extensive study incorporating much larger cohorts that may ultimately utilize this procedure as a viable tool in cancer diagnosis and treatments.

## Figures and Tables

**Figure 1 mps-02-00086-f001:**
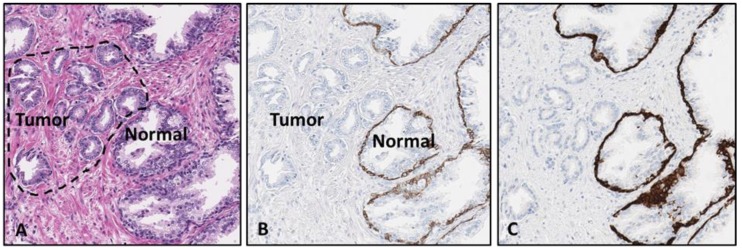
Malignant primary prostate adenocarcinoma tissue sample. The first image shows the prostate tissue stained with H&E (**A**). The areas with tumor and normal prostate gland tissue are labeled. The H&E stained tissue slide was de-stained and anti-CK5/14 mouse monoclonal antibody cocktail was applied to determine feasibility of the proposed protocol (**B**). A sequential sample slide was stained with the same anti-CK5/14 marker using a standard protocol procedure (right panel), for comparison of stain intensity to initial de-stain/re-stain procedure results (**C**). [10× magnification].

**Figure 2 mps-02-00086-f002:**
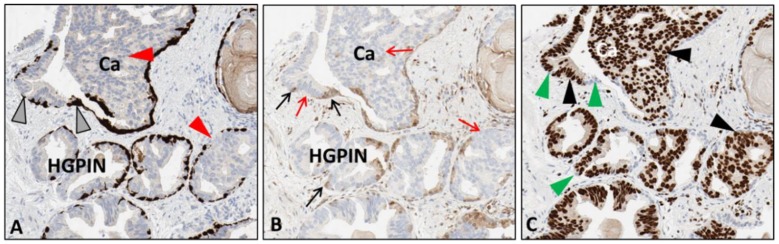
Sequential slides of human prostate tissue exhibiting cancer (Ca) invading into normal glands and high-grade prostatic intraepithelial neoplasia (HGPIN). There is an aggressive carcinoma invading glandular structures that have retained normal basal cells. The retained basal cells are positive for HMWCK + p63 (**A**) (grey arrowheads), positive for phosphatase tensin homolog (PTEN) (**B**) (black arrows), and negative for ETS-Related Gene (ERG) (**C**) (green arrowheads). The cancer is negative for HMWCK + p63 (**A**) (red arrowheads), PTEN (**B**) (red arrows), but positive for ERG (**C**) (black arrowheads). [10× magnification].

**Figure 3 mps-02-00086-f003:**
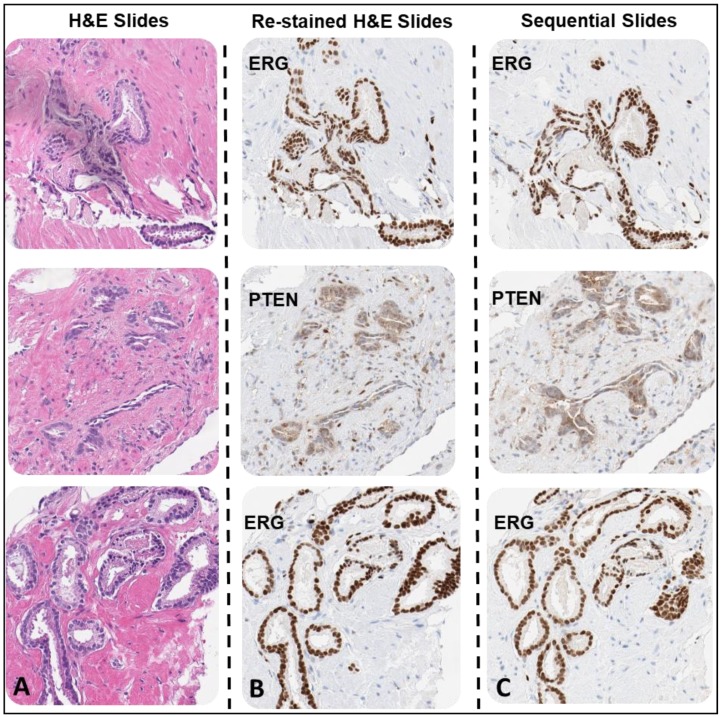
Prostate cancer core needle biopsies (CNBs) sample H&E slides with PTEN and ERG IHC DAB stained slides. The first slide for each sample was H&E-stained (**A**). Each sample H&E was de-stained and re-stained with either PTEN or ERG antibody depending on the pathologist analysis for biomarker loss or positivity to demonstrate tumor heterogeneity (**B**). The additional sequential slides for each sample were stained with anti-PTEN antibody or anti-ERG antibody (**C**). Each sample H&E was de-stained and re-stained with either PTEN or ERG antibody depending on the pathologist analysis for biomarker loss or positivity to demonstrate tumor heterogeneity (10× magnification).

**Figure 4 mps-02-00086-f004:**
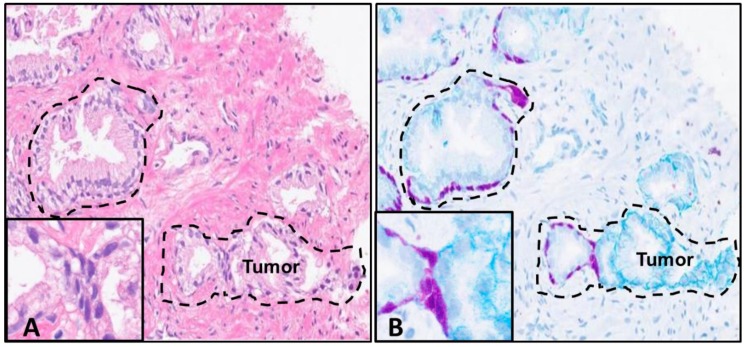
Prostate adenocarcinoma sample CNB H&E and Chromogen IHC. The initial H&E-stained slide with the dotted line indicating prostatic intraepithelial neoplasia (PIN) lesion and tumor area (left panel) (**A**). The de-stained H&E that was stained with HMWCK + p63 mouse monoclonal antibody cocktail and anti-CD49f rabbit polyclonal antibodies using Dual Chromogen detection (right panel) (**B**). The anti-HMWCK +p63 antibody cocktail (purple) stains the basal cells of normal prostatic glands, and the anti-CD49f (teal) antibody stains normal basal cell membranes (masked by the HMWCK +p63) but demonstrates an intracellular and cytoplasmic expression in aggressive tumors (area demonstrating budding tumor outlined in H&E-stained slide in the left panel and the right panel) (20× magnification with 60× instep]).

**Figure 5 mps-02-00086-f005:**
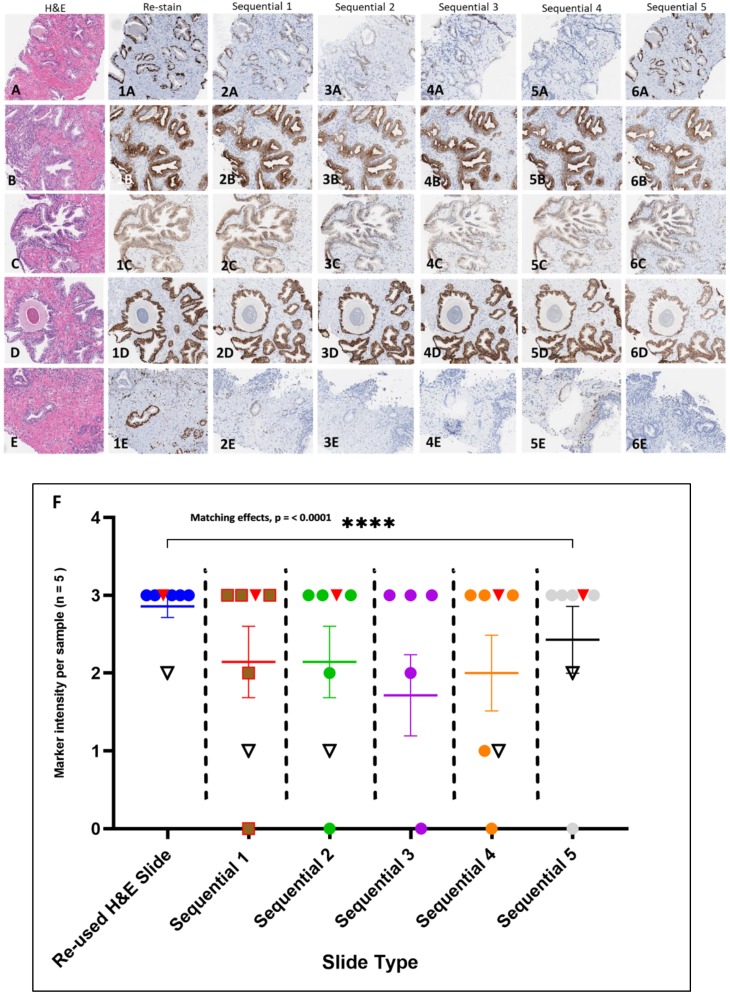
H&E stained slide image and reused H&E slide selected antibody immunostain comparison with sequential slides in PCa CNBs. (**A** through **6E**). H&E stained Slides (**A**–**E**). Antibodies: HMWCK + p63 (**1A**–**5A**) (Note: uneven data points for Sequential slide 5 due to lack of immunostain for HMWCK + p63); CK 8 &18 (**1B**–**6B**); CD49f (**1C**–**6C**); E-cadherin (**1D**–**6D**); ERG (**1E**–**6E**). Scatter plot assessment of side by side comparison of pathologist analysis scores and comments for sequential slide and reused H&E antibody immunostaining. Note: the red inverted triangles represent the ERG internal controls and the open inverted triangles with black outline represent the CD49f internal controls (**F**). Data presented as SEM with Chi square, df (18.12, 1). The results determined by RM one-way ANOVA matching across rows (see Table 4) showing significant matching with *p* value (< 0.0001). The analysis was performed using GraphPad Prism version 8.2.1. [Images 4× magnification].

**Figure 6 mps-02-00086-f006:**
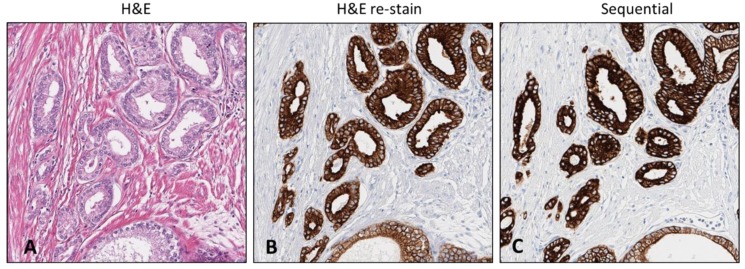
The H&E stained slide archived 4+ years subjected to de-stain and re-stain procedure compared to the sequential sample slide. Initial H&E stained slide containing region of tumor (**A**). CK 8 &18 antibody re-stained slide retaining region of interest and exact architecture (**B**). Sequential slide comparison immunostained with CK 8 &18 exhibiting comparable stain intensity (**C**). [10× magnification].

**Table 1 mps-02-00086-t001:** List of initially assess sample H&E stained slides with specimen parameters, de-stain results and antibodies tested.

Tissue	Specimen	H&E De-Stain Result	Antibodies Used	Initial H&E Analysis
Prostate	Resection	+	p40	Malignant Primary_Adenocarcinoma, Gleason 3 + 3 = 6
Prostate	Resection ^a^	+	CK5/14: PTEN	NA
Prostate	Resection ^a^	+	CK5/14: p504s	Malignant Primary_Adenocarcinoma, Gleason 3 + 3 = 6
Liver (Pancreas Met) *	CNB ^a^	+	PTEN/p504s/CK5/14	Mock CNBs (due to the cut)
Liver	Resection	+/-	HMWCK + p63	NA
Lung	Resection	+/-	HMWCK + p63	NA
N. colon	Resection	+	HMWCK + p63	NA
Skin	Resection	-	E-cadherin	NA
Skin	Resection	-	E-cadherin	NA
Prostate	TMA	+	ERG	NA
Prostate	CNB	+	CD49f	3+3
Prostate	CNB	+	CD49f	no cancer
Prostate	CNB	+	CD49f	no cancer
Prostate	CNB	+	PTEN/ERG ^b^	High grade growing into normal glands 3+3
Prostate	CNB	+	PTEN/ERG ^b^	little bit of tumor grade 3+3
Prostate	CNB	+	PTEN/ERG ^b^	3+3, area of tumor, fragmented tumor
Prostate	CNB	+	PTEN/ERG ^b^	no tumor
Prostate	CNB	+	PTEN/ERG ^b^	grade 4 and 5 cancer, High grade
Prostate	CNB	+	PTEN/ERG ^b^	low grade 3, Lot of PIN
Prostate	CNB	+	PTEN/ERG ^b^	atrophy, inflammation
Prostate	CNB	+	PTEN/ERG ^b^	no cancer, small nerve area
Prostate	CNB	+	PTEN/ERG ^b^	a little fragment tumor
Prostate	CNB	+	PTEN/ERG ^b^	HGPIN, few basal cells left, some cancer
Prostate	CNB	+	PTEN/ERG ^b^	Atypical adenomatous hyperplasia and atrophic glands, Central zone lesion
Prostate	CNB	+	PTEN/ERG ^b^	1 mm grade 3 tumor
Prostate	CNB	+	PTEN/ERG ^b^	small tumor area
Prostate	CNB	+	PTEN/ERG ^b^	small tumor area
Prostate	CNB	+	CD49f/HMWCK ^b^	high grade cancer 4 and 5 trying to make glands invading into norm glands
Prostate	CNB	+	CD49f/HMWCK ^b^	grade 5 cancer (high grade), PIN
Prostate	CNB	+	CD49f/HMWCK ^b^	small amount of tumor no basal cells
Prostate	Resection	+	CK 8 &18	NA
Prostate	CNB	+	CD49f	little bit of tumor grade 3+3
Prostate	CNB	+	HMWCK + p63	grade 4 and 5 cancer, High grade
Prostate	CNB	+	p504s	low grade 3, Lot of PIN
Prostate	CNB	+	CK 8 &18	cancer (3 + 3 with normal)
Prostate	CNB	+	PTEN	3 + 3 ERG positive tumor, folded over
Prostate	CNB	+	CD49f	3 + 3 lesion: 3 cores
Prostate	CNB	+	ERG	3 + 3 involving 2/2 cores
Prostate	CNB	+	CD49f	2 cores: 3 + 3 involving 2/2 cores
Prostate	CNB	+	CK 8 & 18	3 + 3 lesion in one frag 1mm heterogeneous chromatin
Prostate	CNB	+	HMWCK + p63	tumor 3 + 3 Atrophic glands, edge normal
Prostate	CNB	+	HMWCK + p63	no tumor
Prostate	CNB	+	HMWCK + p63	Atrophy, inflammation
Prostate	CNB	+	CK 8&18	atrophy, inflammation
Prostate	CNB	+	CD49f	no cancer
Prostate	CNB	+	ERG	3 + 3 fragmented tumor lost basal cells

Abbreviations: Met, Metastasis; CNB, Core needle biopsy: NA, Not applicable: ^a^ Multiple H&Es prepared. ^b^ Dual chromogen immunostaining. * Mock needle cores: Appropriate H&E de-stain, (+); Moderate de-stain, (+/-); retention of H&E, (-).

**Table 2 mps-02-00086-t002:** Time tracking and coverslip parameters.

Sample Archive Time	Coverslip Removal Time	Type of Coverslip
1 month	10 min	Thin film
1 year	10 min	Thin film
2 year	10 min–60 min	Thin film
4 year	~38 h	Thin film
4+ year **	~47 h	Thin film
2+ year #	1–2 days	Glass
5 year	4–5 days	Glass
5 year	4–5 days	Glass
12 year	4–5 days	Glass

Abbreviations: h, hours; ** Sample Archived 4 years 11 months; # Samples Archived 2 years 11 months ~Approximation due to time at removal.

**Table 3 mps-02-00086-t003:** Immunohistochemistry (IHC) Antibodies and Adapted Staining Protocols.

*Antibody (clone)*	*HMWCK + p63 (34*β*E12)*	*p504s (SP116)*	*CK 8 &18 (B22.1 &B23.1)*	*PTEN (SP218)*	*E-cadherin (36)*	*CD49f*	*ERG (EPR3864)*
Species	mouse monoclonal	rabbit monoclonal	mouse monoclonal	rabbit monoclonal	mouse monoclonal	rabbit polyclonal	rabbit monoclonal
Antibody Vendor	Ventana Medical Systems, Inc., Tucson, Arizona	Cell-Marque, Rocklin, California	Ventana Medical Systems, Inc., Tucson, Arizona			N/A	Ventana Medical Systems, Inc., Tucson, Arizona
**De-stained H&E slide Immunohistochemistry adapted protocol**							
IHC platform	Ventana Benchmark ULTRA						
Detection Kit	Ventana OptiView DAB IHC		Ventana *ultra*View DAB IHC	Ventana OptiView DAB IHC			
Deparaffin	none						
HIER	64 min CC1 (pH 8.5)	32 min CC1 (pH 8.5)	36 min CC1 (pH 8.5)	56 min CC1 (pH 8.5)	63 min CC1 (pH 8.5)	64 min CC1 (pH 8.5)	32 min CC1 (pH 8.5)
Blocking	Peroxidase block						
Ab incubation parameters	36 °C, 16 min	36 °C, 32 min	37 °C, 16 min	37 °C, 16 min	36 °C, 24 min	36 °C, 24 min	36 °C, 32 min

Abbreviations: Ab, antibody; CC, cell conditioning; DAB, 3, 3’- diaminobenzidine HIER, heat-induced epitope retrieval; min, minutes.

**Table 4 mps-02-00086-t004:** Comparison of H&E re-used and sequential comparator immunostaining intensity scores.

Marker	Re-used H&E Stain Intensity	Sequential Stain Intensity: 0–3 (int ctrl)					Initial H&E Assessment
HMWCK + p63	3	2	2	2	0	3	atrophy inflammation
CK8&18	3	3	3	3	3	3	Atrophy + inflammation: whole glands
CD49f	3 (2)	3 (1)	3 (1)	3 (1)	3 (1)	3 (2)	no cancer, int ctrls
E-cadherin	3	3	3	3	3	3	not much cancer but weird well differentiated
ERG	3 (3)	0 (3)	0 (3)	0 (0)	1 (3)	0 (3)	lots of infiltrating lymphocytes, grade 3 + 3 fragmented tumor lost basal cells, int ctrls

Abbreviation: int ctrls, internal controls.

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
