# Peer review of "A Method to Reuse Archived H&E Stained Histology Slides for a Multiplex Protein Biomarker Analysis"

_mps, 2019, doi:10.3390/mps2040086_

Round 1
Reviewer 1 Report
The manuscript by Hinton et al presents a protocol for reprocessing historical H&E slides for IHC. The resulting slides where then stained with several antibodies and shown to give very comparable results to fresh sections cut from the same blocks. The protocol is not entirely novel, as extracting excess eosin with alcohol, and hematoxylin with acidic solutions is common knowledge in histology, however validating that this process does not interfere with antigen labeling for a number of antibodies (and thus presumably for a great many IHC stains) is valuable. Furthermore, knowing that stains can in principle be extracted is distinct from actually working out the most reliable way to do it.
I have some minor questions:
1)
“The reused 5-year archived H&E stained slides (skin 317 resections) sealed with glass coverslips also required extended time for coverslip removal and 318 reagent rinses. The resulting H&E stain removal exhibited residual H&E stain on the slides resulting 319 in incomplete immunostaining (ERG) (data not shown).”
It isn’t obvious to me why older slides should have different results when extracting the H&E stains as compared to newer slides. The use of a glass coverslip might have some effect, but apparently other glass coverslip specimens could be processed. Is this effect actually related to the age of the specimen, or did the lab possibly change how the slides were prepared (different stain composition)?
2)
“Reaction Buffer (Sodium Citrate) (Proprietary reagent, Ventana/RTD, Tucson, AZ, USA)”
This is vague given how important removing the excess hematoxylin is (the process fails without it). Generally acidic solutions can be used to remove hematoxylin. If this reagent is really proprietary and no information is available its composition, the pH should be given so that it can be compared to conventional solutions (e.g. acid alcohol).
Aside from these points, I recommend publication.
Author Response
We thank you for your astute attention given to this procedure and have given a point by point response in detail for your questions and observations below:
Point 1: “The reused 5-year archived H&E stained slides (skin 317 resections) sealed with glass coverslips also required extended time for coverslip removal and 318 reagent rinses. The resulting H&E stain removal exhibited residual H&E stain on the slides resulting 319 in incomplete immunostaining (ERG) (data not shown).”
It isn’t obvious to me why older slides should have different results when extracting the H&E stains as compared to newer slides. The use of a glass coverslip might have some effect, but apparently other glass coverslip specimens could be processed. Is this effect actually related to the age of the specimen, or did the lab possibly change how the slides were prepared (different stain composition)?
Response 1: We have added the following text to line 322-326 on page 12: We have found that the storage conditions of older archived H&E stained slides (particularly with glass coverslips) causes extensive adhesion of the coverslip to the tissue slide due to the extended time in storage, requiring a slight extension of extraction procedures. Also, we observed that pre-analytics will impact the H&E removal resulting in some residual retention”. Unfortunately, due to the age of the slide, pre-analytical data was not available.
Point 2: “Reaction Buffer (Sodium Citrate) (Proprietary reagent, Ventana/RTD, Tucson, AZ, USA)”
This is vague given how important removing the excess hematoxylin is (the process fails without it). Generally acidic solutions can be used to remove hematoxylin. If this reagent is really proprietary and no information is available its composition, the pH should be given so that it can be compared to conventional solutions (e.g. acid alcohol).
Response 2: You are correct, generally acidic solutions are applied for hematoxylin removal. We have included the following correction to line 111 page 4 : (Tris based, 7.6 ± 0.2 pH).
Reviewer 2 Report
The manuscript titled "A Method to Reuse Archived H&E Stained Histology Slides for a Multiplex Protein Biomarker Analysis." provides in-depth technique for re-utilization of H&E slides in multiplex protein analysis. The manuscript is well written and provides comprehensive details for the procedure.
Author Response
We thank you for your attention given to this procedure and for the consideration of the manuscript for publication.